# Treatment effect modification due to comorbidity: Individual participant data meta-analyses of 120 randomised controlled trials

Peter Hanlon[1], Elaine W. Butterly[1], Anoop SV Shah[2], Laurie J. Hannigan[3,4,5], Jim Lewsey[1], Frances S. Mair[1], David M. Kent[6], Bruce Guthrie[7], Sarah H. Wild[7], Nicky J. Welton[4], Sofia Dias[8], David A. McAllister[1]*

1 School for Health and Wellbeing, University of Glasgow, Glasgow, United Kingdom, 2 London School of Hygiene and Tropical Medicine, London, United Kingdom, 3 Nic Waals Institute, Lovisenberg Diaconal Hospital, Oslo, Norway, 4 Population Health Sciences, Bristol Medical School, University of Bristol, Bristol, United Kingdom, 5 Department of Mental Disorders, Norwegian Institute of Public Health, Olso, Norway, 6 Predictive Analytics and Comparative Effectiveness Center, Tufts Medical Center/Tufts University School of Medicine, Boston, Massachusetts, United States of America, 7 Usher Institute, University of Edinburgh, Edinburgh, United Kingdom, 8 Centre for Reviews and Dissemination, University of York, York, United Kingdom

* David.mcallister@glasgow.ac.uk

## Abstract

### Background

People with comorbidities are underrepresented in clinical trials. Empirical estimates of treatment effect modification by comorbidity are lacking, leading to uncertainty in treatment recommendations. We aimed to produce estimates of treatment effect modification by comorbidity using individual participant data (IPD).

### Methods and findings

We obtained IPD for 120 industry-sponsored phase 3/4 trials across 22 index conditions ($n$ = 128,331). Trials had to be registered between 1990 and 2017 and have recruited ≥300 people. Included trials were multicentre and international. For each index condition, we analysed the outcome most frequently reported in the included trials. We performed a two-stage IPD meta-analysis to estimate modification of treatment effect by comorbidity. First, for each trial, we modelled the interaction between comorbidity and treatment arm adjusted for age and sex. Second, for each treatment within each index condition, we meta-analysed the comorbidity–treatment interaction terms from each trial. We estimated the effect of comorbidity measured in 3 ways: (i) the number of comorbidities (in addition to the index condition); (ii) presence or absence of the 6 commonest comorbid diseases for each index condition; and (iii) using continuous markers of underlying conditions (e.g., estimated glomerular filtration rate (eGFR)). Treatment effects were modelled on the usual scale for the type of outcome (absolute scale for numerical outcomes, relative scale for binary outcomes). Mean age in the trials ranged from 37.1 (allergic rhinitis trials) to 73.0 (dementia trials) and percentage of male participants range from 4.4% (osteoporosis trials) to 100%

**Data Availability Statement:** Aggregated data and code required to run these models, along with full model descriptions, are available at https://doi.org/10.5281/zenodo.7713360. Individual participant data is available upon application to the trial sponsors (via https://www.

clinicalstudydatarequest.com/ or https://yoda.yale.edu/), subject to a data transfer agreement.

**Funding:** This work was funded by the Wellcome Trust (grant number 201492/Z/16/Z, grant recipients DMA, SD) and the Medical Research Council (grant number MR/S021949/1, grant recipient PH). The funders had no role in study design, data collection and analysis, decision to publish, or preparation of the manuscript.

**Competing interests:** I have read the journal's policy and the authors of this manuscript have the following competing interests: SD received fees from the Association of the British Pharmaceutical Industry (ABPI) for delivery of a Masterclass (unrelated to this work). The other authors have declared that no competing interests exist.

**Abbreviations:** ARR, absolute risk reduction; BASFI, Bath Ankylosing Spondylitis Functional Index; BASDI, Bath Ankylosing Spondylitis Disease Activity Index; CI, credible interval; COMET, Core Outcome Measures in Effectiveness Trials; COPD, chronic obstructive pulmonary disease; CSDR, Clinical Study Data Request; DAPT, dual antiplatelet therapy; eGFR, estimated glomerular filtration rate; IPD, individual participant data; MBP, mid-blood pressure; MCID, minimum clinically important difference; MDRD, Modification of Diet in Renal Disease; MedDRA, Medical Dictionary for Regulatory Activities; RCT, randomised controlled trial; SGLT2, sodium-glucose co-transporter-2; YODA, Yale Open Data Access.

(benign prostatic hypertrophy trials). The percentage of participants with 3 or more comorbidities ranged from 2.3% (allergic rhinitis trials) to 57% (systemic lupus erythematosus trials). We found no evidence of modification of treatment efficacy by comorbidity, for any of the 3 measures of comorbidity. This was the case for 20 conditions for which the outcome variable was continuous (e.g., change in glycosylated haemoglobin in diabetes) and for 3 conditions in which the outcomes were discrete events (e.g., number of headaches in migraine). Although all were null, estimates of treatment effect modification were more precise in some cases (e.g., sodium-glucose co-transporter-2 (SGLT2) inhibitors for type 2 diabetes—interaction term for comorbidity count 0.004, 95% CI −0.01 to 0.02) while for others credible intervals were wide (e.g., corticosteroids for asthma—interaction term −0.22, 95% CI −1.07 to 0.54). The main limitation is that these trials were not designed or powered to assess variation in treatment effect by comorbidity, and relatively few trial participants had >3 comorbidities.

## Conclusions

Assessments of treatment effect modification rarely consider comorbidity. Our findings demonstrate that for trials included in this analysis, there was no empirical evidence of treatment effect modification by comorbidity. The standard assumption used in evidence syntheses is that efficacy is constant across subgroups, although this is often criticised. Our findings suggest that for modest levels of comorbidities, this assumption is reasonable. Thus, trial efficacy findings can be combined with data on natural history and competing risks to assess the likely overall benefit of treatments in the context of comorbidity.

## Author summary

### Why was this study done?

- There is often uncertainty about how treatments for single conditions should be applied to people with 2 or more long-term conditions (multimorbidity).

- People with multimorbidity are underrepresented in randomised controlled trials (RCTs); however, trials rarely report whether the efficacy of treatment differs by the number of additional long-term conditions (comorbidities) or in the presence of specific comorbidities.

### What did the researchers do and find?

- We analysed individual-participant data from 120 RCTs including 128,331 participants across 23 index conditions to assess whether the efficacy of treatment differed depending on the number of comorbidities or in the presence of any of the most common comorbidities.

- We found no evidence that treatment efficacy differed depending on the number of comorbidities, or by any specific comorbidities, for any of the index conditions and treatment comparisons included in this analysis.

**What do these findings mean?**

- Within the range of comorbidities included within these trials, treatment effects did not vary by comorbidity. These findings can be used within evidence syntheses to estimate the likely overall benefit of treatments in the context of multimorbidity.

- These findings are limited by the fact that people with multiple comorbidities are under-represented in trials, and those with the highest degree of comorbidity are often excluded.

## Introduction

Multimorbidity, the presence of 2 or more long-term conditions, is a global clinical and public health priority [1,2]. Most people with a given long-term condition also have comorbidities (referring to additional long-term conditions in the context of an index condition). There is uncertainty about how individual long-term conditions should be managed in the presence of comorbidities [3]. A major driver of this uncertainty is the underrepresentation of people with multimorbidity in randomised controlled trials (RCTs) [4,5]. Trial populations are typically younger, healthier, and have fewer comorbidities than people treated in routine clinical practice. This has led clinical guideline developers to caution against the application of single-disease recommendations for people with multimorbidity [6]. However, despite the challenges to clinical management posed by this uncertainty, the efficacy of treatments in the context of comorbidity is rarely assessed. It is therefore not clear, for most treatments, whether relative treatment efficacy differs in people with comorbidity.

Assessing individual differences in response to medical treatments is a controversial topic. Differences in treatment efficacy are typically assessed using subgroup analyses. Subgroup analyses in RCTs seek to assess if treatment efficacy differs by patient characteristics [7]. Testing of prespecified subgroup effects is common practice in RCTs of medical therapies [8,9]. As such, subgroup analyses seek to inform stratified approaches to patient care by identifying groups for whom recommendations may be tailored [10]. However, trials rarely report subgroup analyses by levels of comorbidity or for specific comorbidities. Furthermore, subgroup analyses are inconsistently executed and reported, as well as suffering a number of well-documented statistical pitfalls [7,11], notably that analysis of subgroups risks false positive and false negative findings [11]. RCTs are generally not powered to detect subgroup effects, and as such, the sample size in subgroup analyses is frequently insufficient to detect clinically significant differences in treatment efficacy even if these were to exist [12]. Conversely, by testing multiple subgroups, the likelihood of chance findings (i.e., false positives) is increased [7,12].

The limitations of trial-level subgroup analyses can be reduced using meta-analyses. However, when considering whether treatment efficacy varies by comorbidity, traditional study-level meta-analysis of published findings are likely to be inadequate as trials rarely report subgroup effects by comorbidity, and those that do may be subject to publication bias. In such circumstances, any assessment of the effect of comorbidity is therefore based on between-trial comparisons that are prone to bias [13]. Individual-participant data meta-analysis has the potential to overcome these problems. We previously demonstrated, using data from >100 industry-sponsored clinical trials, that it was possible to identify comorbidities in most trials and that multimorbidity was common (although underrepresented) in trial populations [4,14,15]. Furthermore, in a recent simulation study, we demonstrated that combining trials

on all comparisons for a given indication in Bayesian hierarchical models has several desirable properties in terms of estimating treatment effect modification by comorbidity [16]. First, precision is higher compared to single-comparison meta-analyses, increasing the likelihood of detecting small (but clinically relevant) subgroup effects where these are present. Secondly, extreme values are attenuated towards the null (shrinkage), reducing the risk of false positive findings [16]. Bayesian hierarchical models may therefore be a useful tool to assess treatment efficacy estimates in the context of multimorbidity.

This study aims to assess whether treatment effects are modified in the presence of comorbidity, by using individual participant data (IPD) from 120 trials to assess whether treatment efficacy for 23 index conditions differs by (i) number of additional long-term conditions (comorbidity count); (ii) the 6 commonest comorbidities for each index condition; (iii) by continuous biomarkers associated with comorbidity.

## Methods

### Study design

For trials of 23 index conditions, we identified comorbid long-term conditions using IPD for each trial. We then summarised these as a comorbidity count (in addition to the index condition) for each participant. Further, we identified the 6 commonest comorbidities for each index condition across trials and defined a presence/absence variable for each. We estimated differences in treatment efficacy by fitting regression models to IPD for each trial to obtain trial-level estimates of covariate–treatment interaction effects. We fit models for age and sex alone, for a comorbidity count, and for each of the 6 commonest comorbidities for each index condition. Trial-level estimates were then meta-analysed to obtain drug and index condition-specific estimates of treatment effect modification by comorbidity. This process is summarised in Fig 1 and explained in detail below.

All analyses were conducted in R (R Core Team, 2021). Analysis code, metadata (indicating, for example, how treatment arms and outcomes were selected), and data (except trial IPD) are available on the project github repository (https://doi.org/10.5281/zenodo.7713360).

### Data sources

Trials were identified according to a prespecified protocol [17]. We focused on trials of pharmacological agents for 23 index conditions (Table 1). Eligibility criteria were RCTs for one of the index conditions; registered with the United States Clinical trials registry (clinicaltrials.gov) on or after January 1990; phase 2/3, 3, or 4; including ≥300 participants; and with eligibility defined using an upper age limit of 60 years or more or no upper age limit. Smaller studies and studies with lower age limits were excluded as they were considered less likely to include sufficient people with comorbidity. From a list of all registered, eligible trials we then identified trials for which IPD were available from one of 2 repositories: Clinical Study Data Request (CSDR) or the Yale Open Data Access (YODA) repository. These repositories facilitate sharing of industry-sponsored trial data with third-party researchers. The process of trial identification is described in detail elsewhere [4].

### Quantifying comorbidity

For each participant with a specified index condition in each of the included trials, we identified comorbidities from a prespecified list of 21 conditions (cardiovascular disease, chronic pain, arthritis, affective disorders, acid-related disorders, asthma/chronic obstructive pulmonary disease (COPD), diabetes mellitus, osteoporosis, thyroid disease, thromboembolic

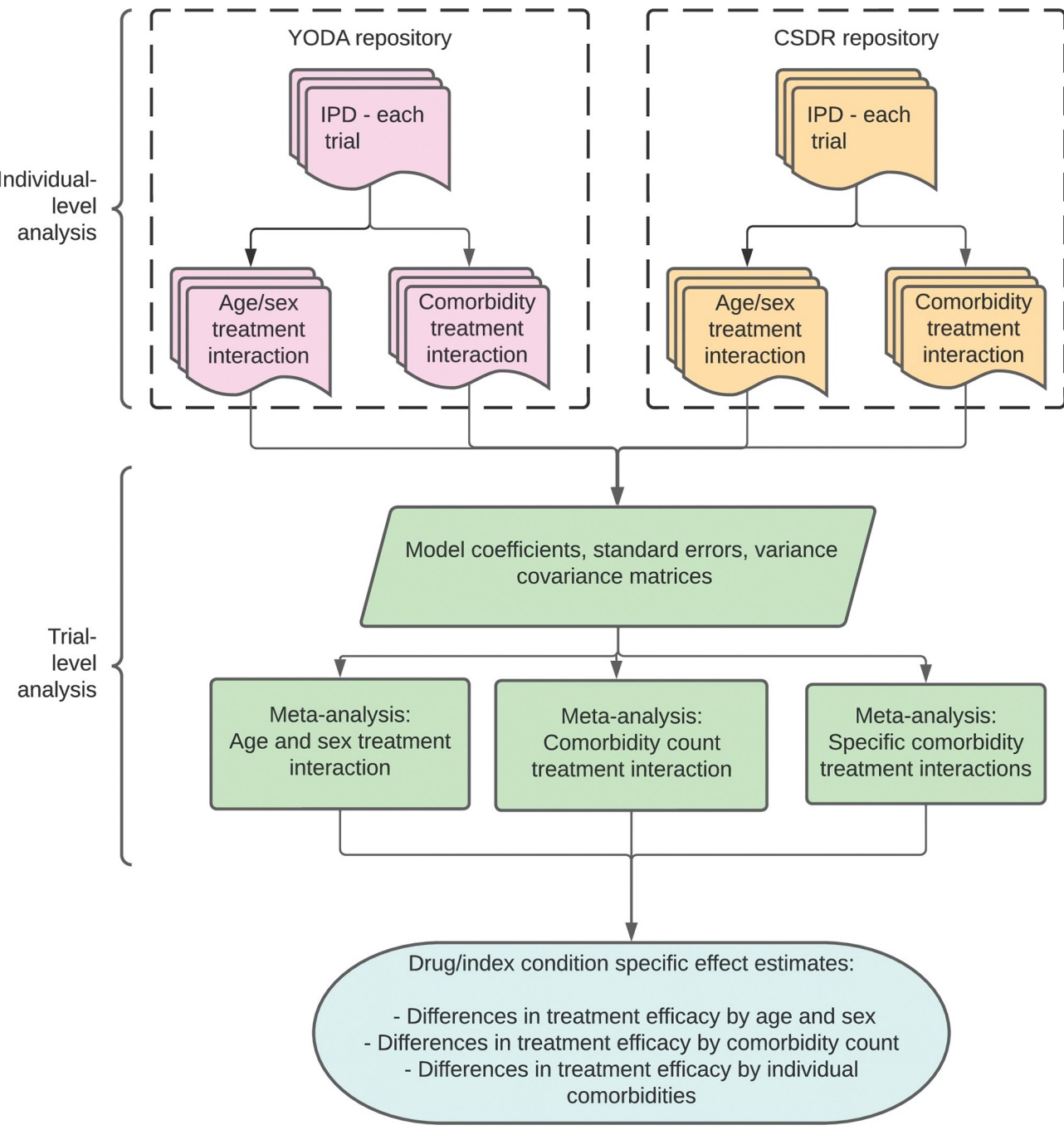

**Fig 1. Overview of analysis.** This figure gives an overview of the analysis structure and hierarchy. Analyses of individual-level data within each trial were conducted within 2 secure repositories (YODA shown in pink and CSDR shown in orange). For each trial, a summary of the results was exported and meta-analysed within each treatment indication (green). CSDR, Clinical Study Data Request; YODA, Yale Open Data Access.

disease, inflammatory conditions, benign prostatic hyperplasia, gout, glaucoma, urinary incontinence, erectile dysfunction, psychotic disorders, epilepsy, migraine, parkinsonism, and dementia) [4]. These comorbidities were based on previous work identifying comorbidities within trial IPD and were based on assessment of medical history and concomitant medication data. In this previous work, we demonstrated that while for many trials medical history had

**Table 1. Index conditions, outcomes, and treatment comparisons for included trials.**

| Index conditions | Outcome | Treatment comparisons | Trials | Mean age (SD) | % Male | Comorbidities | | | | |
|---|---|---|---|---|---|---|---|---|---|---|
| | | | | | | Mean | Zero | One | Two | Three or more |
| Trials with continuous outcomes | | | | | | | | | | |
| Ankylosing Spondylitis | BASDAI score | Interleukin inhibitors (L04AC)-IL6 [1], Tumour necrosis factor (TNF-) alpha inhibitors (L04AB) [2] | 3 | 39.9 (11.8) | 66.50% | 1.45 | 23.50% | 34% | 24.70% | 17.80% |
| Asthma | FEV1 | Glucocorticoids (R03BA) [2], Other systemic drugs for obstructive airway diseases (R03DX) [1], Selective beta-2-adrenoreceptor agonists (R03AC) [1] | 4 | 43.3 (16.8) | 40.20% | 0.65 | 52.20% | 33.90% | 11% | 2.90% |
| Benign Prostatic Hypertrophy | IPSS Total Score | Drugs used in erectile dysfunction (G04BE) [5] | 5 | 63.4 (8.5) | 100% | 1.43 | 23.90% | 34.20% | 24.50% | 17.40% |
| Chronic Idiopathic Urticaria | DLQI Score | Other systemic drugs for obstructive airway diseases (R03DX) [3] | 3 | 42.3 (14.1) | 26.50% | 1.59 | 20.40% | 32.40% | 25.80% | 21.40% |
| Dementia | ADAS Score | Anticholinesterases (N06DA) [3] and Thiazolidinediones (A10BG) [3] | 6 | 73 (8.8) | 40.90% | 1.83 | 16% | 29.40% | 26.90% | 27.70% |
| Diabetes | HBA1c | Biguanides (A10BA) vs. Glucagon-like peptide-1 (GLP-1) analogues (A10BJ) [1], Dipeptidyl peptidase 4 (DPP-4) inhibitors (A10BH) [1], Dipeptidyl peptidase 4 (DPP-4) inhibitors (A10BH) vs. Glucagon-like peptide-1 (GLP-1) analogues (A10BJ) [1], Dipeptidyl peptidase 4 (DPP-4) inhibitors (A10BH) vs. Sodium-glucose co-transporter 2 (SGLT2) inhibitors (A10BK) [1], Glucagon-like peptide-1 (GLP-1) analogues (A10BJ) [2], Insulins and analogues (A10A) vs. Glucagon-like peptide-1 (GLP-1) analogues (A10BJ) [2], Sodium-glucose co-transporter 2 (SGLT2) inhibitors (A10BK) [12], Sulfonylureas (A10BB) vs. Dipeptidyl peptidase 4 (DPP-4) inhibitors (A10BH) [1], Sulfonylureas (A10BB) vs. Sodium-glucose co-transporter 2 (SGLT2) inhibitors (A10BK) [1] | 22 | 58.7 (10) | 58.60% | 1.68 | 18.60% | 31.30% | 26.30% | 23.80% |
| Erectile Dysfunction | IPSS Total Score | Drugs used in erectile dysfunction (G04BE) [1] | 1 | 62.1 (8.1) | 100% | 1.24 | 28.90% | 35.90% | 22.20% | 13% |
| Gastro-oesophageal Reflux Disease | Percent heartburn free days | Proton pump inhibitors (A02BC) [2] | 2 | 47.7 (13.5) | 35.10% | 0.62 | 53.8% | 33.4% | 10.3% | 2.5% |
| Gout | Urate | Preparations inhibiting uric acid production (M04AA) [1] | 1 | 51.6 (12.2) | 93.80% | 0.5 | 60.7% | 30.3% | 7.6% | 1.4% |
| Hypertension | Systolic blood pressure | ACE inhibitors, plain (C09AA) vs. Angiotensin II antagonists, plain (C09CA) [3], Angiotensin II antagonists, plain (C09CA) [1], Thiazides and plain (C03AA) [1] | 5 | 58.3 (11.8) | 54.40% | 1.03 | 35.70% | 36.80% | 18.90% | 8.60% |
| Inflammatory Bowel Disease | CDAI Score or MAYO Score | Interleukin inhibitors (L04AC)-IL12-IL23 [3], Selective immunosuppressants (L04AA) [2]. Tumour necrosis factor alpha (TNF-) inhibitors (L04AB) [6] | 11 | 38.5 (12.7) | 50.10% | 0.94 | 39.10% | 36.70% | 17.30% | 6.90% |

**Table 1.** (Continued)

| Index conditions | Outcome | Treatment comparisons | Trials | Mean age (SD) | % Male | Comorbidities | | | | |
|---|---|---|---|---|---|---|---|---|---|---|
| | | | | | | Mean | Zero | One | Two | Three or more |
| Inflammatory Arthropathy | ACR numerical | Interleukin inhibitors (L04AC)-IL12-IL23 [1], Interleukin inhibitors (L04AC)-IL6 [4], Tumour necrosis factor alpha (TNF-) inhibitors (L04AB) [8], Tumour necrosis factor alpha (TNF-) inhibitors (L04AB) vs. Interleukin inhibitors (L04AC)-IL6 [1] | 14 | 50.8 (12.4) | 23.10% | 1.25 | 28.70% | 35.80% | 22.40% | 13.10% |
| Osteoporosis | BMD Total Hip | Bisphosphonates (M05BA) [2], Bisphosphonates (M05BA) vs. Parathyroid hormones and analogues (H05AA) [1], Parathyroid hormones and analogues (H05AA) [2] | 5 | 72 (8.1) | 4.70% | 2.79 | 6.10% | 17.10% | 23.90% | 52.90% |
| Parkinson Disease | UPDRS Total | Dopamine agonists (N04BC) [4] | 4 | 62.7 (10) | 59% | 2.04 | 13% | 26.50% | 27.10% | 33.40% |
| Psoriasis | PASI Score | Interleukin inhibitors (L04AC)-IL12-IL23 [2], Interleukin inhibitors (L04AC)-IL17A [4] | 6 | 45.8 (12.5) | 66.50% | 1.15 | 31.70% | 36.40% | 20.90% | 11% |
| Pulmonary Disease, Chronic Obstructive | FEV1 | Glucocorticoids (R03BA) [3], Selective beta-2-adrenoreceptor agonists (R03AC) [1], Selective beta-2-adrenoreceptor agonists (R03AC) vs. Anticholinergics (R03BB) [2] | 6 | 63.7 (8.5) | 67% | 1.5 | 22.30% | 33.50% | 25.10% | 19.10% |
| Pulmonary Fibrosis | FVC | Other protein kinase inhibitors (L01EX) [2] | 2 | 66.8 (8) | 79.30% | 2.75 | 6.40% | 17.60% | 24.20% | 51.80% |
| Restless Legs Syndrome | RLS Symptom Score Total | Dopamine agonists (N04BC) [3] | 3 | 53.4 (12.9) | 38.90% | 1.85 | 15.70% | 29.10% | 26.90% | 28.30% |
| Rhinitis, allergic | Total Nasal Symptom Score | Fluticasone (R01AD08) [1] | 1 | 37.1 (16.4) | 43.40% | 0.59 | 55.40% | 32.70% | 9.60% | 2.30% |
| Systemic Lupus Erythematosus | SLE Disease Activity Index | Selective immunosuppressants (L04AA) [2] | 2 | 37.8 (11.5) | 6% | 2.99 | 5% | 15% | 22.50% | 57.50% |
| Trials with categorical outcomes | | | | | | | | | | |
| Migraine | No. headaches | Topiramate (N03AX11) [5] | 5 | 39.3 (11.9) | 14.60% | 0.75 | 47.20% | 35.40% | 13.30% | 4.10% |
| Osteoporosis | Vertebral fracture | Parathyroid hormones and analogues (H05AA) [1] and Bisphosphonates (M05BA) [2] | 3 | 72.8 (7) | 4.40% | 2.66 | 7% | 18.60% | 24.70% | 49.70% |
| Thromboembolic | Bleeding [1]; DVT or PE [1]; DVT or PE and Bleeding [7] | Vitamin K antagonists (B01AA) vs. Direct thrombin inhibitors (B01AE) [4], Heparin group (B01AB) [1], Heparin group (B01AB) vs Direct thrombin inhibitors (B01AE) [3], Direct thrombin inhibitors (B01AE) [1] | 9 | 64.9 (13.2) | 58.30% | 1.57 | 20.80% | 32.70% | 25.60% | 20.90% |

"Treatment comparisons" indicates the treatment comparisons for each trial based on drug class using the WHO ATC code (for L04AC-Interleukin inhibitors, the ATC class was also further split according to the specific interleukin(s). Where there is only a single code the comparator is either placebo or usual care). Trial-level data on number included, age, sex and comorbidity distributions are shown in Hanlon and colleagues [4], supplementary file 6.

BASDAI, Bath Ankylosing Spondylitis Disease Activity Index; TNF, tumour necrosis factor; FEV1, forced expiratory volume in 1 s; IPSS, international prostate symptom score; DLQI, dermatology life quality index; ADAS, Alzheimer's Disease Assessment Scale; HbA1c, glycated haemoglobin; GLP1, glucagon-like peptide-1; DPP4, dipeptidyl peptidase 4; SGLT2, sodium-glucose co-transporter 2; ACE, angiotensin converting enzyme; CDAI, Crohn's Disease Activity Index; ACR, American College of Rheumatology; BMD, bone mineral density; UPDRS, Unified Parkinson's Disease Rating Scale; PASI, psoriasis area and severity index; FVC, forced vital capacity; RLS, restless legs syndrome; SLE, systemic lupus erythematosus; DVT, deep vein thrombosis; PE, pulmonary embolism.

been redacted, data on concomitant medications were widely available and could be used to define comorbidities [4]. This involved combining some conditions into the same definition (e.g., asthma and COPD, which could not be differentiated based on medication use alone). These definitions were based on the World Health Organisation Anatomic Therapeutic Classification and are described in our previous publication and available on the project github repository [4]. Where medical history data was available and coded using the Medical Dictionary for Regulatory Activities (MedDRA) coding system, we also identified the same conditions using MedDRA codes.

### Comorbidity count

For the primary analysis, we created a comorbidity count for each participant. This was the total number of comorbidities present, not including the index condition. This count was used as a numerical variable in all analyses.

### Individual comorbidities

For each index condition, we also identified the 6 most common comorbidities from the full list of 21 possible comorbidities. These individual comorbidities were analysed as binary variables (reflecting the presence of absence of that specific comorbidity).

### Selected biomarkers/risk factors

In addition to the 21 comorbidities defined using medication and/or medical history, we identified 5 continuous biomarkers that may indicate comorbidity (e.g., renal impairment, hypertension, anaemia, or liver disease) or risk factors (e.g., obesity). These were based on baseline trial measurements: estimated glomerular filtration rate (eGFR, as a marker of renal impairment, taken from trial data where this was available and calculated from creatinine, age, sex, and race using the Modification of Diet in Renal Disease (MDRD) equations if it was not), body mass index (as recorded or calculated based on height and weight), fibrosis-4 (FIB-4) index (as a marker of liver disease calculated from aspartate aminotransferase, alanine transaminase, and platelet counts), haemoglobin, and mid-blood pressure (MBP, defined as $0.5 \times$ (systolic blood pressure + diastolic blood pressure)).

### Demographics

Age and sex were extracted from each trial based on the trial recorded values at randomisation.

### Treatment arms

Treatment arm comparisons were prespecified prior to undertaking the outcome analyses. For multiarm trials, the most extreme arms were selected for comparison (e.g., if different dosages were used, the highest dose was compared to placebo or usual care—e.g., canagliflozin 300 mg, rather than 100 mg, versus placebo). Where placebo or usual care was included as a trial arm, this was selected as the comparator. Otherwise, we chose the arm with the least recently developed treatment as the comparator arm. This was to give the best chance of identifying effect modification, with the resulting analysis representing an upper limit on the degree of effect modification observed.

## Outcomes

We aimed to identify outcomes common across trials to facilitate meta-analysis. We obtained information from clincialtrials.gov via the Database for Aggregated Analysis of ClinicalTrials. gov (AACT; https: //ctti-clinicaltrials.org/citation-policy/) on all outcomes (primary and secondary) for each trial. For each index condition, we then identified 1 or more outcomes that appeared to be common to multiple trials (e.g., forced expiratory volume in 1 s (FEV1) in COPD trials, 6-min walk distance (6MWD) in pulmonary hypertension trials). Within the trial repositories, we then reviewed the trial documentation to identify these outcomes for each trial. For trials of anticoagulants, in addition to the efficacy outcome, we also analysed bleeding events as these are a common and clinically important adverse outcome.

## Statistical analyses

In 4 separate analyses we (i) estimated age and sex–treatment interactions without including comorbidity; (ii) estimated comorbidity–treatment interactions for the comorbidity count; (iii) estimated comorbidity–treatment interactions for the 6 commonest comorbidities for each index condition; and (iv) examined covariate–treatment interactions for continuous biomarkers. Full descriptions of the modelling are provided in the Supporting information appendix (S1 File) and are described briefly below.

## IPD analysis

For trials where the outcome was a continuous variable, for each trial and analysis the change in each outcome was modelled using linear regression. For analysis (i), the final measure was regressed on the baseline measure, age (modelled as a continuous variable scaled to 15-year increments, which was close to the standard deviation for most trials), sex (male versus referent group of females), arm (binary variable treatment/control), and interactions with arm for each covariate. For American College of Rheumatology-N (ACR-N, a measure of improvement in disease activity in rheumatoid arthritis, which is itself a measure of change), we did not include the baseline measure as a covariate. We then repeated this modelling for the remaining analyses (ii to iv) adding comorbidity covariates in addition to age and sex (comorbidity count, specific comorbidities, and continuous biomarkers for analyses (ii to iv), respectively). From these models, the model coefficients, standard errors, and variance-covariance matrices were obtained and exported from the YODA and CSDR secure analysis platforms.

For trials where the outcome was a count or a binary variable, we fitted similar models using Poisson regression and logistic regression, respectively.

## Meta-analysis

For the continuous outcomes, in order to convert the measures onto a similar scale, we divided the estimates and standard errors by the minimum clinically important difference (MCID) for that measure. For most outcome measures, higher scores indicate worse outcomes (e.g., Bath Ankylosing Spondylitis Disease Activity Index (BASDI)). Where this was not the case (e.g., FEV1), we multiplied the values by minus one so that the direction of effect was the same for all trials. For the variance-covariance matrix, we divided each element by the MCID-squared. The MCID was selected using the published literature by hand-searching papers in the Core Outcome Measures in Effectiveness Trials (COMET) database for relevant conditions [18]. This search was supplemented by simple internet searches (Google searches using the full and abbreviated names for each outcome and MCID, MID, "minimum clinically important difference," or "minimum important difference"). Where no published MCID recommendations

could be found, we used the MCID defined in the power calculations in the trial protocols. At this stage, for each index condition, we restricted the analysis to the single most common outcome across trials. In 2 index conditions (Ankylosing spondylitis and hypertension), 2 outcomes were equally common; BASDAI and Bath Ankylosing Spondylitis Functional Index (BASFI) and diastolic blood pressure and systolic blood pressure, we arbitrarily chose BASDAI in the former case and chose systolic blood pressure in the latter as it is more prominent in clinical decision-making.

For each drug class, the model outputs were then meta-analysed. We used random-effects meta-analyses where 5 or more trials were included within the same drug class, and fixed effects where there were fewer than 5 trials. We used Bayesian models since this allowed us to simultaneously model multiple coefficients (e.g., age–treatment and sex–treatment interactions). The Bayesian models were fit using the *brms* package [19]. Samples from the posterior distribution were obtained and summarised as the mean and 95% credible intervals (CI). *P*-values were not presented as this was a Bayesian analysis. The full posteriors are provided in the project repository (doi:10.5281/zenodo.7713360).

In case other researchers wish to use the results of our models of treatment–covariate interactions to inform subsequent analyses as informative priors, we obtained summaries of the posterior predictions. We did so only for analysis (ii) for continuous outcomes. In order to provide a more general set of priors, we also predicted the comorbidity count–treatment interaction for treatment comparisons/conditions not included in our model by obtaining samples from the posteriors. The latter are provided to allow researchers to conduct Bayesian analyses or probabilistic sensitivity analyses if studying conditions/treatment comparisons not included in our modelling as this represents a prediction for an unobserved index condition/treatment comparison (albeit one which is assumed to be exchangeable with the conditions/treatment comparisons included in the current analysis). We then summarised these samples by fitting a Student's t-distribution. As with the main analysis, these models were fitted using the *brms* package (S1 File).

## Ethical approval

This project had approval from the University of Glasgow, College of Medicine, Veterinary and Life Sciences ethics committee (200160070).

## Results

### Trial characteristics

Trial baseline characteristics have been reported previously [4]. For trials with continuous outcomes, there were 20 index conditions and 47 treatment comparisons across a total of 106 trials ($n$ = 88,150 participants). For 9 index conditions, there was only 1 treatment comparison across all trials. Diabetes, which was the condition for which there were the most trials (22), had the largest number of treatment comparisons (9) (Table 1). Within each model, all trials had a single common outcome except inflammatory bowel disease, where the ulcerative colitis trials used the MAYO score and Crohn's disease trials used the Crohn's Disease Activity Index score. For trials with categorical outcomes, there were 3 index conditions (migraine, osteoporosis, and thromboembolism) and 11 treatment comparisons across a total of 17 trials ($n$ = 11,624 participants). For thromboembolism, there were 3 more specific categories of indication—primary prevention (5 trials), secondary prevention (2 trials), and treatment (2 trials).

## Continuous outcomes—Age–and sex–treatment interactions

For all conditions with continuous outcomes, interaction terms for age–and sex–treatment interactions are shown in Table 2. For most drug classes, interaction terms for age included the null, indicating no statistically significant associations consistent with modification of treatment efficacy by age. However, in the diabetes trials, there appeared to be an attenuation in the treatment effect with increasing age for 3 drug classes (0.07 (95% CI 0.00, 0.13) for sulfonylureas versus SGLT2 inhibitors, 0.09 (95% CI 0.01, 0.17) for DPP-4 inhibitors versus SGLT2 inhibitors, and 0.07 (95% CI 0.04, 0.11) for SGLT2 inhibitors versus placebo). Taking SGLT2 inhibitors versus placebo as an example, this can be read as follows—"the lowering effect on HbA1c of SGLT2 inhibitors versus placebo is 0.28 (95% CI 0.16, 0.44) mmol/mol smaller (since the MCID for HbA1c is 4 mmol/mol) per 15-year increment in age, for age 50 years versus age 80 years, this corresponds to the effect being 0.56 (95% CI 0.32, 0.88) mmol/mol smaller. Similarly, most interaction terms for sex included the null, with a few exceptions (Table 2). For example, for glucagon-like peptide-1 (GLP-1) analogues, the interaction term for sex was 0.29 (0.12, 0.49) indicating that the lowering effect on HbA1c of GLP-1 analogues is 1.16 (0.48, 1.96) mmol/mol smaller in men than in women.

## Continuous outcomes—Comorbidity–treatment interactions

For each drug class, Figs 2 to 6 show the main treatment effect (black points, expressed as change in minimally clinically important difference) and the estimate for the comorbidity–treatment interaction based on a comorbidity count (red points) meta-analysed within treatment indications. Fig 7 shows similar estimates for indications in which only a single trial was included. Meta-analyses for each drug class are shown in Figs 2 to 6 and, for classes where only 1 trial was analysed, trial-level estimates are shown in Fig 7. Comorbidity count was not associated with any attenuation or strengthening in treatment efficacy; in all cases, the 95% CIs included the null. This suggests that for all treatments and in all index conditions, it is plausible that there is no difference in treatment effect by comorbidity (on the absolute scale) within the range of comorbidity counts observed in the trials. When examining comorbidity–treatment interactions for the 6 most common comorbidities within each index condition, 95% CIs included the null for all estimates (S1 Table). Similarly, when assessing modification of treatment efficacy by continuous biomarkers, all estimates included the null (S2 Table).

In a sensitivity analysis, rather than using a fixed effects model for meta-analyses where there were fewer than 5 trials, we used a random effects model. The 95% CIs were wider, but the results of these models were otherwise similar to those presented in the main analysis (S3 Table).

## Informative priors for subsequent analyses including different index condition/treatment comparisons

On predicting treatment effect modification by comorbidity count for a notional unobserved condition and notional unobserved treatment comparison, the samples from the posterior were approximately t-distributed (central estimate = 0.01, dispersion = 0.01, degrees of freedom = 3.24).

## Categorical outcomes—Morbidity count–treatment interactions

For the 3 index conditions with categorical outcomes (Table 1), there was no evidence of any comorbidity count–treatment interactions. These findings are summarised in Table 3.

**Table 2. Covariate–treatment interactions (expressed as multiples of minimal clinically important difference) by age and sex for continuous outcomes; point estimates and 95% CIs.**

| Index condition | Treatment comparison | Age–treatment interaction | Sex–treatment interaction |
|---|---|---|---|
| Ankylosing Spondylitis | Tumour necrosis factor alpha (TNF-) inhibitors (L04AB) | −0.02 (−0.09, 0.05) | −0.06 (−0.19, 0.06) |
| Ankylosing Spondylitis | Interleukin inhibitors (L04AC)-IL6 | −0.01 (−0.10, 0.08) | 0.03 (−0.13, 0.20) |
| Asthma | Selective beta-2-adrenoreceptor agonists (R03AC) | −1.16 (−2.05, −0.28)* | −1.15 (−3.18, 0.87) |
| Asthma | Glucocorticoids (R03BA) | 0.12 (−0.37, 0.60) | −0.40 (−1.41, 0.64) |
| Asthma | Other systemic drugs for obstructive airway diseases (R03DX) | 0.39 (−0.36, 1.13) | −0.84 (−2.25, 0.58) |
| Benign Prostatic Hypertrophy | Drugs used in erectile dysfunction (G04BE) | −0.04 (−0.51, 0.42) | - |
| Chronic Idiopathic Urticaria | Other systemic drugs for obstructive airway diseases (R03DX) | 0.09 (−0.15, 0.33) | 0.58 (0.07, 1.07)* |
| Dementia | Thiazolidinediones (A10BG) | 0.30 (−0.01, 0.61) | 0.19 (−0.15, 0.54) |
| Dementia | Anticholinesterases (N06DA) | 0.09 (−0.15, 0.32) | −0.02 (−0.29, 0.26) |
| Diabetes | INSULINS AND ANALOGUES (A10A) vs. Glucagon-like peptide-1 (GLP-1) analogues (A10BJ) | 0.02 (−0.06, 0.09) | 0.04 (−0.05, 0.13) |
| Diabetes | Biguanides (A10BA) vs. Glucagon-like peptide-1 (GLP-1) analogues (A10BJ) | 0.05 (−0.04, 0.14) | −0.00 (−0.12, 0.12) |
| Diabetes | Sulfonylureas (A10BB) vs. Dipeptidyl peptidase 4 (DPP-4) inhibitors (A10BH) | 0.01 (−0.04, 0.06) | 0.03 (−0.04, 0.10) |
| Diabetes | Sulfonylureas (A10BB) vs. Sodium-glucose co-transporter 2 (SGLT2) inhibitors (A10BK) | 0.07 (0.00, 0.13)* | 0.04 (−0.04, 0.12) |
| Diabetes | Dipeptidyl peptidase 4 (DPP-4) inhibitors (A10BH) vs. Glucagon-like peptide-1 (GLP-1) analogues (A10BJ) | 0.08 (−0.04, 0.21) | −0.02 (−0.17, 0.14) |
| Diabetes | Dipeptidyl peptidase 4 (DPP-4) inhibitors (A10BH) vs. Sodium-glucose co-transporter 2 (SGLT2) inhibitors (A10BK) | 0.09 (0.01, 0.17)* | 0.05 (−0.05, 0.15) |
| Diabetes | Dipeptidyl peptidase 4 (DPP-4) inhibitors (A10BH) | 0.02 (−0.03, 0.06) | −0.01 (−0.07, 0.06) |
| Diabetes | Glucagon-like peptide-1 (GLP-1) analogues (A10BJ) | 0.02 (−0.13, 0.17) | 0.29 (0.12, 0.49)* |
| Diabetes | Sodium-glucose co-transporter 2 (SGLT2) inhibitors (A10BK) | 0.07 (0.04, 0.11)* | −0.01 (−0.05, 0.03) |
| Erectil Dysfunction | Drugs used in erectile dysfunction (G04BE) | 0.33 (−0.34, 0.99) | - |
| Gastro-oesophageal Reflux Disease | Proton pump inhibitors (A02BC) | −0.01 (−0.05, 0.04) | −0.10 (−0.18, −0.01)* |
| Gout | Preparations inhibiting uric acid production (M04AA) | 0.01 (−0.40, 0.43) | −0.51 (−1.78, 0.77) |
| Hypertension | ACE inhibitors, plain (C09AA) vs Angiotensin II antagonists, plain (C09CA) | 0.28 (−0.12, 0.70) | −0.02 (−0.60, 0.59) |
| Hypertension | Thiazides, plain (C03AA) | −0.01 (−0.72, 0.70) | 1.04 (−0.07, 2.15) |
| Hypertension | Angiotensin II antagonists, plain (C09CA) | −0.26 (−1.21, 0.70) | −0.02 (−1.36, 1.33) |
| Inflammatory Bowel Disease | Selective immunosuppressants (L04AA) | 0.11 (−0.04, 0.27) | 0.09 (−0.16, 0.34) |
| Inflammatory Bowel Disease | Tumour necrosis factor alpha (TNF-) inhibitors (L04AB) | 0.06 (−0.09, 0.22) | −0.04 (−0.40, 0.26) |
| Inflammatory Bowel Disease | Interleukin inhibitors (L04AC)-IL12-IL23 | 0.11 (−0.08, 0.30) | 0.05 (−0.27, 0.36) |
| Inflammatory Arthropathy | Tumour necrosis factor alpha (TNF-) inhibitors (L04AB) vs Interleukin inhibitors (L04AC)-IL6 | 0.39 (−1.04, 1.82) | 2.41 (−0.66, 5.48) |
| Inflammatory Arthropathy | Tumour necrosis factor alpha (TNF-) inhibitors (L04AB) | 0.17 (−0.44, 0.79) | −0.87 (−1.82, 0.11) |
| Inflammatory Arthropathy | Interleukin inhibitors (L04AC)-IL12-IL23 | 1.72 (−1.92, 5.36) | 0.31 (−5.40, 6.03) |
| Inflammatory Arthropathy | Interleukin inhibitors (L04AC)-IL6 | 0.28 (−0.33, 0.88) | 0.00 (−1.12, 1.12) |
| Osteoporosis | Bisphosphonates (M05BA) vs. Parathyroid hormones and analogues (H05AA) | 0.06 (−0.05, 0.17) | −0.00 (−0.26, 0.25) |
| Osteoporosis | Parathyroid hormones and analogues (H05AA) | −0.13 (−0.27, 0.02) | - |
| Osteoporosis | Bisphosphonates (M05BA) | 0.01 (−0.05, 0.07) | 0.30 (0.14, 0.45)* |
| Parkinson Disease | Dopamine agonists (N04BC) | 0.19 (−0.20, 0.59) | −0.28 (−0.77, 0.19) |
| Psoriasis | Interleukin inhibitors (L04AC)-IL12-IL23 | 0.04 (−0.11, 0.19) | −0.43 (−0.68, −0.18)* |
| Psoriasis | Interleukin inhibitors (L04AC)-IL17A | 0.07 (−0.03, 0.16) | −0.30 (−0.47, −0.12)* |
| Pulmonary Disease, Chronic Obstructive | Selective beta-2-adrenoreceptor agonists (R03AC) | −0.34 (−0.91, 0.22) | 0.00 (−0.68, 0.68) |
| Pulmonary Disease, Chronic Obstructive | Glucocorticoids (R03BA) | 0.07 (−0.19, 0.32) | −0.15 (−0.43, 0.15) |

*(Continued)*

**Table 2.** (Continued)

| Index condition | Treatment comparison | Age–treatment interaction | Sex–treatment interaction |
|---|---|---|---|
| Pulmonary Disease, Chronic Obstructive | Selective beta-2-adrenoreceptor agonists (R03AC) vs. Anticholinergics (R03BB) | 0.09 (−0.26, 0.42) | 0.12 (−0.27, 0.53) |
| Pulmonary Fibrosis | Other protein kinase inhibitors (L01EX) | 0.05 (−0.48, 0.56) | −0.22 (−0.83, 0.42) |
| Restless Legs Syndrome | Dopamine agonists (N04BC) | −0.05 (−0.45, 0.36) | −0.09 (−0.78, 0.58) |
| Rhinitis, allergic | fluticasone (R01AD08) | −0.86 (−2.55, 0.84) | 1.40 (−2.32, 5.13) |
| Systemic Lupus Erythematosus | Selective immunosuppressants (L04AA) | 0.03 (−0.12, 0.19) | −0.24 (−0.74, 0.26) |

Estimates are expressed as multiples of the minimum clinically important difference for each outcome. The effect estimates (age and sex) were obtained from a model of each outcome on treatment arm, age, sex and age–and sex–treatment interactions. Blank cells in the sex column are where there were no female participants in the trial (benign prostatic hypertrophy and erectile dysfunction) or no male participants (osteoporosis).

* Indicates where the 95% CI does not include the null.

## Discussion

In an IPD meta-analysis of 120 trials, we examined whether the efficacy of drug treatments differed by comorbidity. For 20 index conditions where the outcome variable was continuous (e.g., glycosylated haemoglobin in diabetes trials), efficacy did not differ by the total number of comorbidities or by the presence or absence of specific comorbidities. Similarly, for 3 conditions (17 trials) examining outcomes which were discrete events (e.g., thromboembolism, bleeding, headaches, and fractures), there was no evidence of treatment effect modification by comorbidity count or by specific comorbidities.

Several previous studies have reported findings on treatment effect modification in IPD meta-analyses and meta-analyses of reported subgroup effects. However, these have largely been confined to major cardiovascular disease trials (e.g., for showing similar efficacy of statin in people with and without diabetes [20], differential benefit of blood pressure lowering therapy in people with and without diabetes [21], or showing questionable net benefit of aspirin in primary prevention [22]) or to concordant conditions defined as those closely related to the index condition or target event for the trial (such as hypertension in stroke trials [23]). These studies have not considered the impact of comorbidity more broadly or of discordant comorbidities not related to the index condition of the trial. This represents an important omission, because there are a number of mechanisms by which the presence of discordant conditions might plausibly modify treatment efficacy (positively or negatively) including increased diagnostic misclassification, altered pharmacokinetics, or pharmacodynamics (e.g., altered drug excretion in people with mild renal impairment or increased benefits of antiplatelet drugs in the presence of coexistent inflammatory conditions) and altered treatment-related behaviours (e.g., better or worse treatment adherence due to existing treatment regimens). Our study adds to this sparse literature showing that, on average, treatment effects are similar across different populations within trials (at modest comorbidity counts of 3 or fewer). This supports the standard assumption that treatment effects are similar when generalising from trial to non-trial eligible populations, at least for populations with limited prevalence of comorbidity such as in these trials.

Although we found that treatment efficacy did not differ by comorbidity count, net overall treatment benefits may nonetheless differ in people with differing degrees of comorbidity. This is because differences in the baseline risk (e.g., the absolute risk of the outcome that the treatment is intended to prevent), differences in susceptibility to treatment-related adverse events, differences in competing risks (e.g., absolute risk of mortality from other causes), and

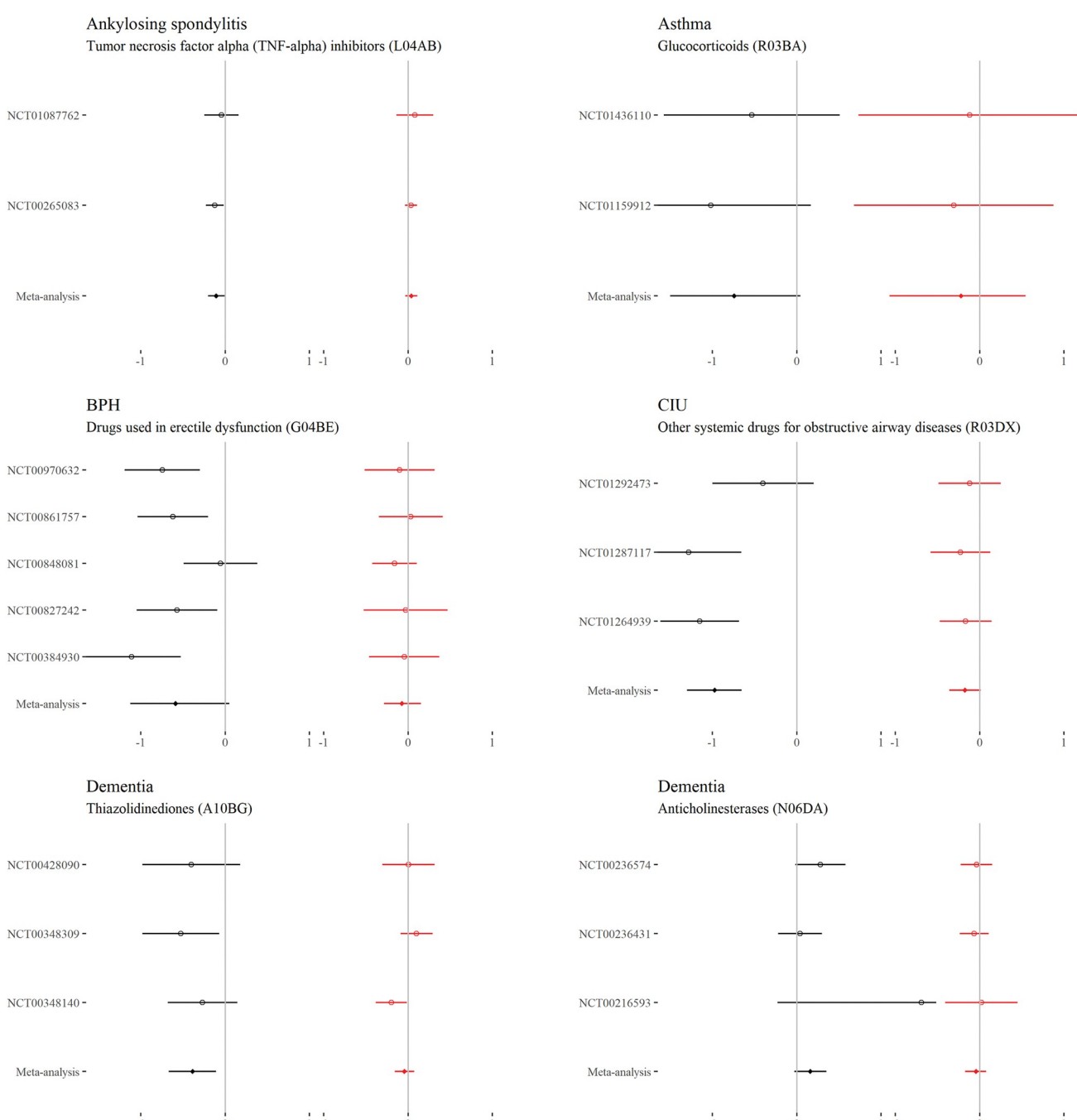

**Fig 2. Main treatment effect and comorbidity–treatment interactions (ankylosing spondylitis, asthma, BPH, CIU, and dementia):** This plot shows the main treatment effect (black) and the comorbidity–treatment interaction (red) based on a comorbidity count. Trial-level estimates (circles) and meta-analysed estimates (diamonds) are presented along with 95% CIs (whiskers). Details of effect estimates, heterogeneity, and model diagnostics can be found here: https://zenodo.org/badge/latestdoi/611754942. BPH, benign prostatic hypertrophy; CI, credibility interval; CIU, chronic idiopathic urticaria.

differences in the burden of treatment (e.g., higher treatment burden in the context of multi-morbidity leading to reduced concordance or reduced quality of life) may all lead to differences in the net overall benefit of treatment [24]. Therefore, where comorbidity alters the (baseline) natural history of diseases, the likelihood of adverse treatment effects (e.g., comorbid

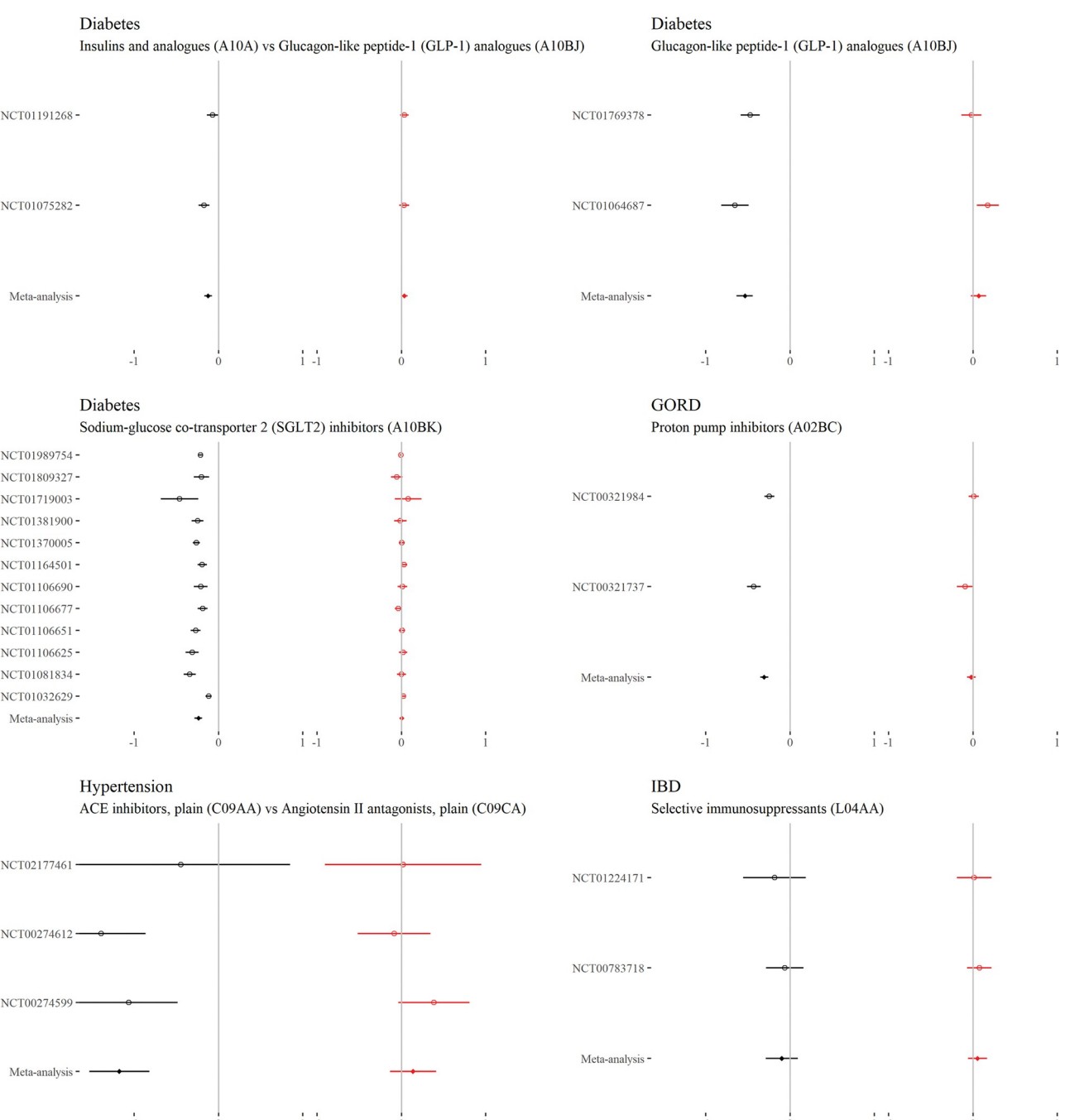

**Fig 3. Main treatment effect and comorbidity–treatment interactions (diabetes, GORD, hypertension, and IBD):** This plot shows the main treatment effect (black) and the comorbidity–treatment interaction (red) based on a comorbidity count. Trial-level estimates (circles) and meta-analysed estimates (diamonds) are presented along with 95% CIs (whiskers). Details of effect [3] estimates, heterogeneity, and model diagnostics can be found here: https://zenodo.org/badge/latestdoi/611754942. CI, credibility interval; GORD, gastro-oesophageal reflux disease; IBD, inflammatory bowel disease.

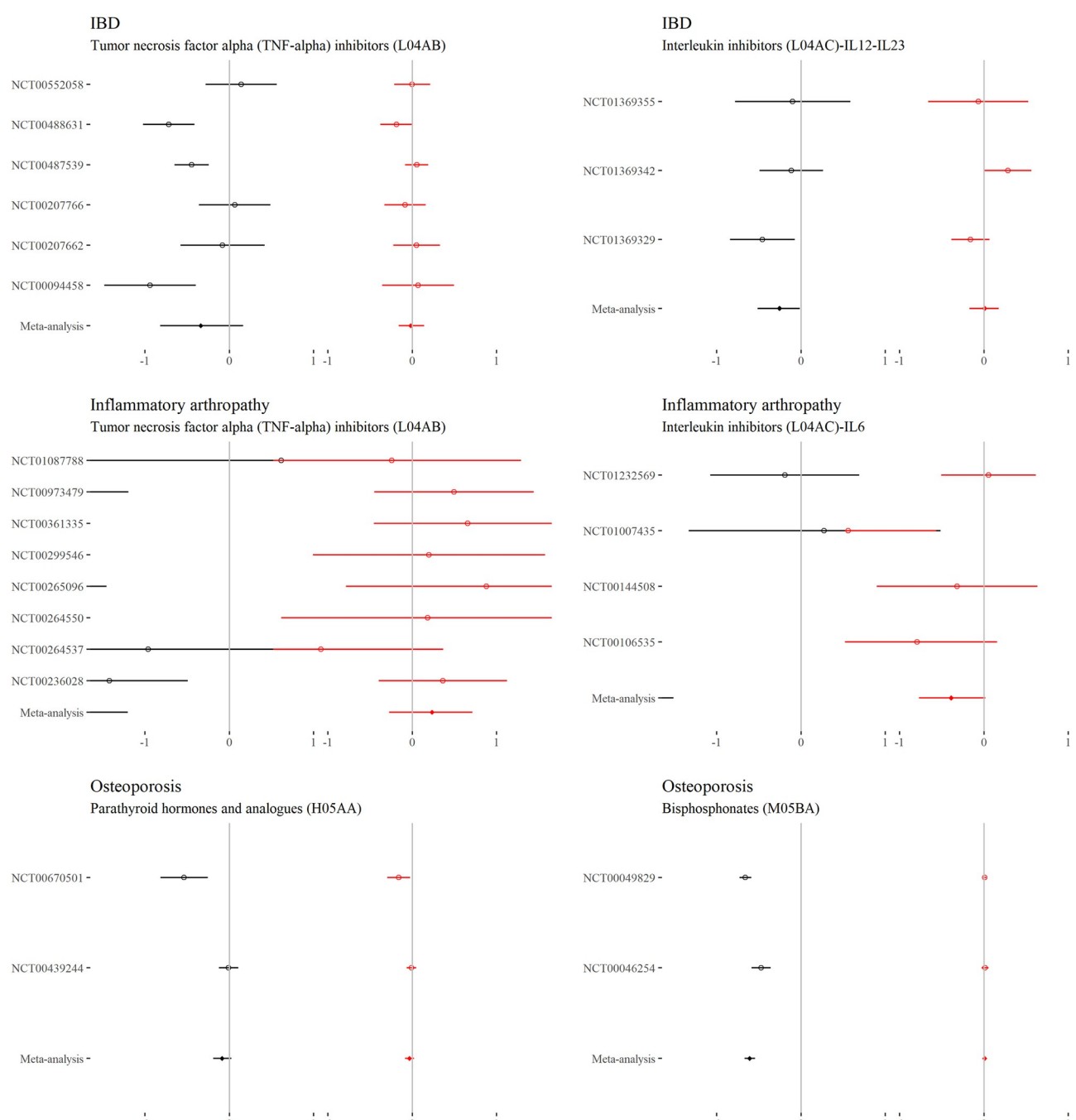

**Fig 4. Main treatment effect and comorbidity–treatment interactions (IBD, inflammatory arthropathy, and osteoporosis):** This plot shows the main treatment effect (black) and the comorbidity–treatment interaction (red) based on a comorbidity count. Trial-level estimates (circles) and meta-analysed estimates (diamonds) are presented along with 95% CIs (whiskers). Details of effect estimates, heterogeneity, and model diagnostics can be found here: [4]https://zenodo.org/badge/latestdoi/611754942. CI, credibility interval; IBD, inflammatory bowel disease.

renal impairment) or life expectancy (e.g., via discordant comorbidities associated with mortality), the effects of treatment must differ even assuming that there is no difference in efficacy. For example, while there is strong evidence that the benefits of dual antiplatelet therapy (DAPT) following myocardial infarction (versus a single antiplatelet) outweigh the risks

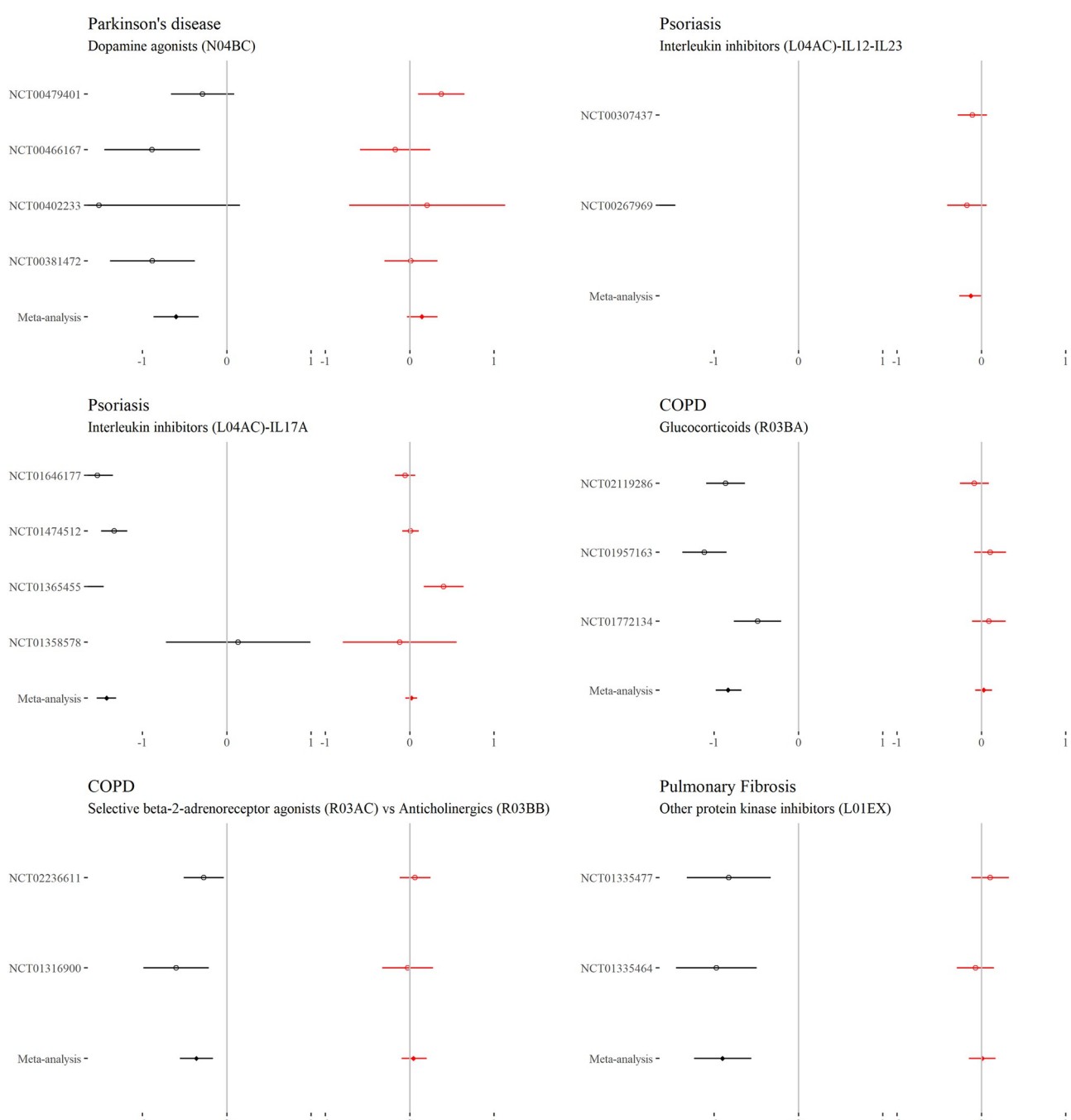

**Fig 5. Main treatment effect and comorbidity–treatment interactions (Parkinson's disease, psoriasis, COPD, and pulmonary fibrosis):** This plot [5]shows the main treatment effect (black) and the comorbidity–treatment interaction (red) based on a comorbidity count. Trial-level estimates (circles) and meta-analysed estimates (diamonds) are presented along with 95% CIs (whiskers). Details of effect estimates, heterogeneity, and model diagnostics can be found here: https://zenodo.org/badge/latestdoi/611754942. CI, credibility interval; COPD, chronic obstructive pulmonary disease.

[25,26], this may not be true for patients with coexisting COPD. Cardiovascular mortality is commoner in COPD than the general population, favouring DAPT [27]. However, non-cardiovascular mortality is also higher [28], favouring single-antiplatelet therapy because of competing risks. Intensive control of blood glucose and other risk factors in diabetes [29,30] and

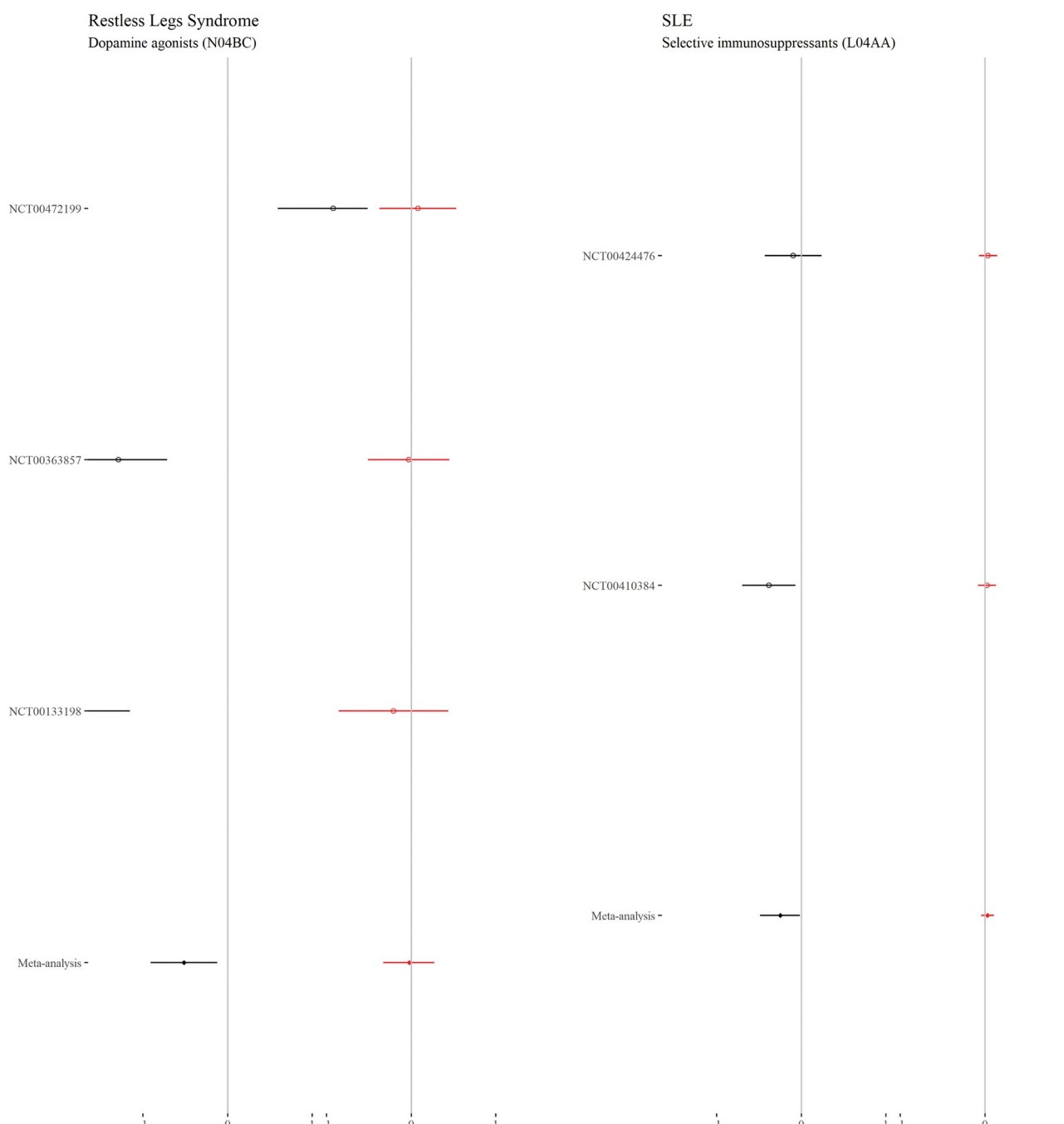

**Fig 6. Main treatment effect and comorbidity–treatment interactions (restless legs syndrome and SLE):** This plot shows the main treatment effect (black) and the comorbidity–treatment interaction (red) based on a comorbidity count. Trial-level estimates (circles) and meta-analysed estimates (diamonds) are presented along with 95% CIs (whiskers). Details of effect estimates, heterogeneity, and model diagnostics can be found here: https://zenodo.org/badge/latestdoi/611754942. CI, credibility interval; SLE, systemic lupus erythematosus.

anticoagulant use in atrial fibrillation [31] provide similar examples where the net overall treatment benefits are uncertain for people with comorbidity.

This is the first IPD clinical trial meta-analysis, as far as we are aware, to examine whether treatment efficacy differs by comorbidity. Nonetheless, there are several important limitations.

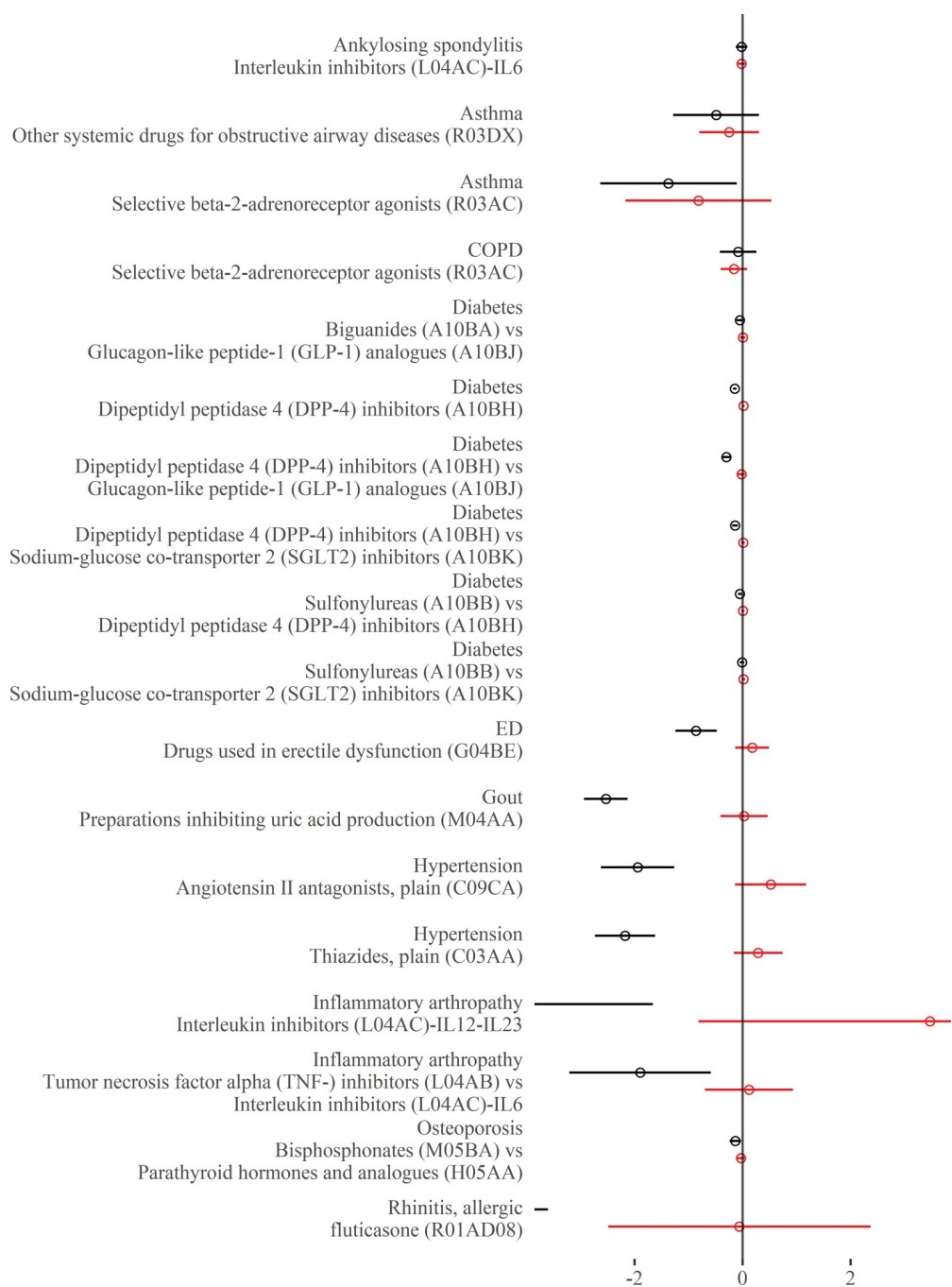

**Fig 7. Main treatment effect and comorbidity–treatment interactions (single trial estimates):** This plot shows the main treatment effect (black) and the comorbidity–treatment interaction (red) based on a comorbidity count. Trial-level estimates (circles) are presented along with 95% CIs (whiskers). Details of effect estimates, heterogeneity, and model diagnostics can be found here: https://zenodo.org/badge/latestdoi/611754942. CI, credibility interval.

First, while for some index conditions (e.g., diabetes) there were many trials, for others there were few trials and so relatively few participants, limiting the precision with which covariate–treatment interactions could be estimated. Furthermore, the individual trials were neither designed not powered to measure comorbidity–treatment interactions. Specifically, the higher

**Table 3. Comorbidity–treatment interactions for binary and count outcomes; point estimates and 95% CIs.**

| Condition | Outcome | Intervention | Comparator | Number of trials | Comorbidity–treatment interaction |
|---|---|---|---|---|---|
| Migraine | No. of headaches | Topiramate (N03AX11) | Placebo | 5 | 1.03 (0.85, 1.22) |
| Osteoporosis | Fracture of vertebrae | Teriparatide (H05AA02) | Placebo | 1 | 0.87 (0.58, 1.30) |
| | | Zoledronic acid (M05BA08) | Placebo | 2 | 0.96 (0.79, 1.17) |
| Primary prevention (thromboembolism) | Bleeding* | Dabigatran (B01AE07) | LMWH (B01AB) | 2 | 1.06 (1.00, 1.12) |
| | | Dabigatran (B01AE07) | Warfarin (B01AA03) | 1 | 1.00 (0.96, 1.04) |
| | DVT or PE | Dabigatran (B01AE07) | LMWH (B01AB) | 3 | 1.07 (0.89, 1.26) |
| Secondary prevention (thromboembolism) | Bleeding* | Dabigatran (B01AE07) | Placebo | 1 | 0.82 (0.64–1.05) |
| | | Dabigatran (B01AE07) | Warfarin (B01AA03) | 1 | 1.06 (0.94–1.19) |
| | DVT or PE | Dabigatran (B01AE07) | Placebo | 1 | 0.96 (0.60–1.54) |
| | | Dabigatran (B01AE07) | Warfarin (B01AA03) | 1 | 1.05 (0.64–1.70) |
| Treatment (thromboembolism) | Bleeding* | Dabigatran (B01AE07) | Warfarin (B01AA03) | 2 | 1.00 (0.90–1.10) |
| | DVT or PE | Dabigatran (B01AE07) | Warfarin (B01AA03) | 2 | 0.91 (0.75–1.10) |

The interaction estimates represent the ratio per one-unit increase in comorbidity count; effect measures estimates above one indicate worse outcomes in the intervention compared to the comparison arm. For number of migraines, the effect estimates are on the rate ratio scale, for the remainder the effect estimates are on the odds ratio scale.

* For anticoagulant medication, where bleeding is a common complication, we analysed bleeding outcomes in addition to efficacy outcomes.

DVT, deep vein thrombosis; LMWH, low molecular-weight heparin; PE, pulmonary embolism.

levels of comorbidity observed in clinical practice (e.g., 5 or more comorbidities) are rare within the trial participants. This reduces the likelihood of detecting a comorbidity–treatment interaction were one to exist. Therefore, while our results are consistent with there being no comorbidity–treatment interactions, this should be interpreted within the range of comorbidities, index conditions, and treatment comparisons that are presented.

Second, most trials were phase 3 trials focussed on efficacy outcomes (e.g., change in a disease marker such as blood pressure or glycosylated haemoglobin) rather than pragmatic trials focussed on harder outcomes (such as the incidence of specific adverse health outcomes). The findings for the smaller number of trials (17 in total) where we did have harder outcomes (headaches, bleeding, thromboembolism, and fracture) were similar to the findings for the remaining trials; there was no evidence of treatment effect modification by comorbidity count on the conventional scale (additive for continuous outcomes and relative for noncontinuous outcomes). Nonetheless, the small number of trials and indications where hard outcomes were studied means that caution is needed in extrapolating our findings to trials or meta-analyses focussing on such outcomes. Also, for some conditions and indications, the main effects were small or included the null. Where this was the case, the chances of detecting treatment effect modification are lower.

Third, while this analysis assesses treatment efficacy, we did not assess whether comorbidities lead to variation in adverse effects of treatment. An appreciation of both benefits and harms is required in order to inform judgements about the net benefits of treatment in the context of comorbidity.

Fourth, while we include a large number of trials across a range of index conditions, this is not a representative sample in terms of the larger body of trials. Specifically, there were all industry-sponsored trials (as the CSDR and YODA repositories only held industry-sponsored trial data for the conditions of interest). Furthermore, not all sponsors share data in this way nor do sponsors share data for all trials conducted. We have previously demonstrated that these trials were similar to the wider body of industry-sponsored trials in terms of characteristics such as size, phase, and significance of the primary outcome [4]. However, it is possible that by selecting only industry-sponsored trials inclusion criteria and selection processes may be more restrictive than for other trials. This means that, while we did not detect any evidence of treatment effect modification by comorbidity, it cannot be assumed to be absent particularly in other trials which may be more pragmatic or have less restrictive selection criteria.

Finally, while comorbidity was present in all the included trials, they remain underrepresentative in terms of the extent comorbidity [4,32–34]. Specifically, there were few people in the included trials with high comorbidity counts (e.g., 4 or more conditions). This highly multimorbid population is not uncommon in routine clinical practice [35] and presents considerable challenges for clinical decision-making [3]. Their exclusion from these trials means that our findings cannot be assumed to be directly transferable to patient groups with the highest degree of multimorbidity, for whom uncertainties over the net benefit of treatments are often greatest [36,37].

Our findings have implications for the conduct of future evidence syntheses. In order to estimate net overall treatment benefits, clinical guidelines and health technology assessments routinely use evidence synthesis [38]. Such approaches combine (i) estimates of relative treatment efficacy with (ii) "natural" history (standard comparator rates) to calculate absolute effectiveness, commonly expressed as the absolute risk reduction (ARR) or number needed to treat [39]. However, hitherto evidence synthesis has rarely been used to estimate net overall treatment effects for people with multimorbidity. This may partly be due to uncertainty as to whether and how efficacy estimates differ in people with and without comorbidities. Since estimating the natural history rates of target and adverse events for people with multimorbidity is relatively straightforward using routine healthcare data (since such data are sufficiently large and rich in people with multimorbidity to produce such estimates), and within the limitations outlined above, our findings support the standard assumption of estimates of treatment efficacy being constant (at least at the modest levels observed within trial populations).

To support such evidence syntheses, we have provided a set of informative priors that can be used to propagate, into the final treatment effectiveness estimates, the additional uncertainty arising from applying estimates from clinical trials to populations rich in multimorbidity. We summarised the variation in treatment effects by comorbidity count as a set of Student's t-distributions. These distributions can be used to inform modelling studies (e.g., health technology assessments) designed to extrapolate treatment effect estimates from trial populations to routine clinical practice where multimorbidity is more common. This has the potential to better inform regulatory bodies and guideline developers as they seek to make treatment recommendations for people with multimorbidity.

Our findings also have relevance for analyses of comorbidity subgroup findings in both single clinical trials and as part of meta-analyses. The lack of information for estimating subgroup effects in clinical trials and dangers of falsely claiming spurious subgroup effects is well established and a range of approaches have been advocated for dealing with this problem. These include limiting the number of subgroups and performing corrections for multiple testing (e.g., the Bonferroni technique used in frequentist analysis), the analysis of treatment effect modification according to participant's prognostic risk scores at baseline (which reduce the dimensionality of the problem and prioritises characteristics known to predict differences in

the rates of target events) [40] and in a Bayesian context the use of subject-matter expert knowledge (via prior elicitation). The prior distributions derived from our modelling for the comorbidity–treatment interactions can help inform such prior-elicitation exercises. Another technique used in Bayesian subgroup analyses is to use off-the-shelf conservative priors designed to avoid over fitting [41]; our findings will help provide reassurance that such priors are unlikely to be overly conservative for modelling comorbidity–treatment interactions.

Finally, our results have relevance for reporting of clinical trial results. Both comorbidity and frailty can be measured using data already collected from clinical trials and—as we show —it is feasible to estimate comorbidity–treatment interactions using such measures. In our project, this required access to IPD a process which is expensive (in terms of analysis time) and complex (requiring formal contractual agreements). The PATH statement advocated that clinical trials should report treatment effect modification by baseline prognostic risk score [40]. We agree that this is a useful approach because it reduces the complex problem of sub-group analysis into a single measure (reducing overfitting), and because, by definition, it targets variables which most strongly predict the risk of target events. This latter aspect is important as it helps inform evidence synthesis models applying trial results to a target population with a higher target event rate. For similar reasons, we propose that trials should also report evidence of treatment effect modification by comorbidity or degree of frailty; this would reduce the risk of overfitting by reducing comorbidity to a single variable that predicts rates of competing events. To inform judgements about net benefits, this same information should be provided for adverse events. In addition, more research is required to establish whether specific comorbidities may attenuate or strengthen treatment efficacy, as if these effects were in the opposite direction for different comorbidities, then a cumulative count of comorbidities may obscure this effect.

In conclusion, we found no evidence that treatment efficacy differed by comorbidity within the levels of comorbidity observed within clinical trial populations. This finding held whether comorbidity was measured using a simple condition count or by the presence or absence of 6 common conditions. Nonetheless, comorbidity is underrepresented in trials, especially at higher levels often seen in clinical practice, and in these contexts, the applicability of trial effect estimates needs to be carefully considered. The analysis of these trials may be used to inform subsequent evidence syntheses, analysis and reporting of individual trials, meta-analyses, and health economic models. We provide model outputs in the form of prior distributions to support such analyses.

## Supporting information

**S1 File. Statistical methods.** This file contains a more detailed description of the statistical analysis and model specifications.
(DOCX)

**S1 Table. Comorbidity–treatment interactions for the 6 most common comorbidities within each index condition.** This table shows the coefficients and 95% CIs for the comorbidity–treatment interaction for each of the 6 most common comorbidities within each index condition. Each comorbidity was modelled separately.
(XLSX)

**S2 Table. Comorbidity–treatment interactions for continuous biomarkers.** This table shows the coefficients and 95% CIs for the comorbidity–treatment interaction continuous biomarkers associated with comorbidity. Biomarkers assessed were estimated glomerular filtration rate (eGFR, as a marker of renal impairment, taken from trial data where this was

available and calculated from creatinine, age, sex, and race using the MDRD equations if it was not), body mass index (as recorded or calculated based on height and weight), fibrosis-4 (FIB-4) index (as a marker of liver disease calculated from aspartate aminotransferase, alanine transaminase, and platelet counts), haemoglobin, and MBP (defined as 0.5 × (systolic blood pressure + diastolic blood pressure)).
(XLSX)

**S3 Table. Random effects meta-analyses for indications with <4 trials.** This table presents the results for meta-analyses of less than 5 trials using a random effects meta-analysis presenting alongside the fixed-effects findings from the main manuscript.
(XLSX)

## Acknowledgments

This study, carried out under YODA Project # 2017–1746, used data obtained from the Yale University Open Data Access Project, which has an agreement with JANSSEN RESEARCH & DEVELOPMENT, L.L.C. The interpretation and reporting of research using this data are solely the responsibility of the authors and does not necessarily represent the official views of the Yale University Open Data Access Project or JANSSEN RESEARCH & DEVELOPMENT, L.L.C. This study was also carried out under ClinicalStudyDataRequest.com project number 1732, used data from the ClinicalStudyDataRequest.com repository, who provided data from Boehringer-Ingelheim, GSK, Lilly, Roche, Takeda, and Sanofi. The interpretation and reporting of research using these data are solely the responsibility of the authors and does not necessarily represent the official views of ClinicalStudyDataRequest.com or Boehringer-Ingelheim, GSK, Lilly, Roche, Takeda or Sanofi.

## Author Contributions

**Conceptualization:** Peter Hanlon, David A. McAllister.

**Data curation:** Peter Hanlon, Elaine W. Butterly, David A. McAllister.

**Formal analysis:** Peter Hanlon, David A. McAllister.

**Funding acquisition:** Peter Hanlon.

**Investigation:** Peter Hanlon.

**Methodology:** Jim Lewsey, David A. McAllister.

**Supervision:** David A. McAllister.

**Visualization:** Peter Hanlon.

**Writing – original draft:** Peter Hanlon.

**Writing – review & editing:** Elaine W. Butterly, Anoop SV Shah, Laurie J. Hannigan, Jim Lewsey, Frances S. Mair, David M. Kent, Bruce Guthrie, Sarah H. Wild, Nicky J. Welton, Sofia Dias, David A. McAllister.

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
