## [Editor Report · Decision Letter 0]

18 Jan 2023

Dear Dr Hanlon, 

Thank you for submitting your manuscript entitled "Assessing treatment effect modification due to comorbidity using individual participant data from industry-sponsored drug trials" for consideration by PLOS Medicine.

Your manuscript has now been evaluated by the PLOS Medicine editorial staff and I am writing to let you know that we would like to send your submission out for external peer review.

Please re-submit your manuscript within two working days, i.e. by Jan 20 2023 11:59PM.

Kind regards,

Philippa Dodd, MBBS MRCP PhD

PLOS Medicine

---

## [Decision Letter · Decision Letter 1]

27 Feb 2023

Dear Dr. Hanlon,

Thank you very much for submitting your manuscript "Assessing treatment effect modification due to comorbidity using individual participant data from industry-sponsored drug trials" (PMEDICINE-D-23-00048R1) for consideration at PLOS Medicine. 

Your paper was evaluated by a senior editor and discussed among all the editors here. It was also sent to independent reviewers, including a statistical reviewer. The reviews are appended at the bottom of this email and any accompanying reviewer attachments can be seen via the link below:

[LINK]

In light of these reviews, I am afraid that we will not be able to accept the manuscript for publication in the journal in its current form, but we would like to consider a revised version that addresses the reviewers' and editors' comments. Obviously we cannot make any decision about publication until we have seen the revised manuscript and your response, and we plan to seek re-review by one or more of the reviewers. 

We expect to receive your revised manuscript by Mar 20 2023 11:59PM. Please email us (plosmedicine@plos.org) if you have any questions or concerns.

We look forward to receiving your revised manuscript. 

Sincerely,

Philippa Dodd, MBBS MRCP PhD

PLOS Medicine

plosmedicine.org

GENERAL

Please respond to all editor and reviewer requests detailed below in full.

Please include line numbers in your revised version starting at 1 and in continuous sequence thereafter.

TITLE

Please revise your title according to PLOS Medicine's style. Your title must be nondeclarative and not a question. It should begin with main concept if possible. "Effect of" should be used only if causality can be inferred, i.e., for an RCT. Please place the study design ("A randomized controlled trial," "A retrospective study," "A modelling study," etc.) in the subtitle (ie, after a colon).

ABSTRACT

Please structure your abstract using the PLOS Medicine headings (Background, Methods and Findings, Conclusions).

Please combine the Methods and Findings sections into one section, “Methods and findings”.

Abstract Methods and Findings:

Please include additional details regarding the study population - number of participants, age ranges etc, and the setting – region/country etc., years during which the studies took place, length of follow up, and main outcome measures.

Paragraph 1: “…markers of underlying conditions (e.g., estimated glomerular function).” Do you mean estimated glomerular filtration rate (eGFR) to monitor renal impairment? (glomerular “function” can be measured in different ways – filtration rate or urinary protein excretion)

Some statements might benefit from further quantification with numerical values and statistical information, for example, paragraph 2 line 1 – how many is “few” 

Paragraph 2 line 2 – as above, would it be helpful to support this statement with statistical information

Paragraph 2 line 6 – Please revise to “Sodium-glucose co-transporter-2 (SGLT2) inhibitors”

PLOS medicine requires that the main results are quantified with 95% CIs as well as p values. When reporting p values please report as p<0.001 or where higher as p=0.002 (not <.001 or p=.002). If not including p values, for the purpose of transparent data reporting, please clearly state the reasons why not.

Please ensure that any/all numbers presented in the abstract are present and identical to those presented in the main manuscript text.

Please include any important dependent variables that are adjusted for in the analyses

In the last sentence of the Abstract Methods and Findings section, please describe the main limitation(s) of the study's methodology.

Abstract Conclusions:

Please emphasize what is new without overstating your conclusions

“Our findings support the assumption that estimates of treatment efficacy are constant…” it would be helpful to include a brief justification/explanation of how this helps/what it adds in real terms

AUTHOR SUMMARY

At this stage, we ask that you include a short, non-technical Author Summary of your research to make findings accessible to a wide audience that includes both scientists and non-scientists. The Author Summary should immediately follow the Abstract in your revised manuscript. This text is subject to editorial change and should be distinct from the scientific abstract. 

We suggest reviewing our website and published articles for examples https://journals.plos.org/plosmedicine/

Please see our author guidelines for more information: https://journals.plos.org/plosmedicine/s/revising-your-manuscript#loc-author-summary

INTRODUCTION

Paragraph 1 sentence 2: suggest removing “The prevalence of multimorbidity is such that” and starting the sentence with “Most people…”

Please see reviewer 2 comments also

METHODS and RESULTS

Please include additional regarding the trials and the participants, for example - participants/sample size and dates of recruitment of participants to the trials, length of follow-up, comorbidities. See also reviewer 3 comments

Page 8 para 2: “…brms package.(29)…” what does this numerical refer to?

PLOS medicine requires that the main results are quantified with 95% CIs as well as p values. When reporting p values please report as p<0.001 or where higher as p=0.002 (not <.001 or p=.002). If not including p values, for the purpose of transparent data reporting, please clearly state the reasons why not.

FIGURES

Please ensure that all figures are associated with an appropriate caption which clearly describes the figure content without the need to refer to the text. Please ensure all abbreviations are defined within (e.g., Figure 1 - CSDR, YODA, IPD)

Figure 2a, b, c – the plots are very small and not easy to read, including the text. Please revise accordingly. Please ensure that an appropriate legend is included which clearly details what the different color bars and dots mean. Please define all abbreviations (e.g., IBD). Please see reviewer 1 comments also.

TABLES

Please ensure that all tables are associated with an appropriate caption which clearly describes the table contents without the need to refer to the text. Please ensure that all abbreviations used in the tables are defined in the caption. For example, 2nd column (outcomes) in table 1, 2nd column table 2 “ACE”

Table 2: would be more accessible if the CIs were placed on the same row. Please revise. Suggest reducing the size of column 2 and considering the use of commas to separate upper and lower bounds in place of the word “to”

Table 2 legend/caption: refers the reader to table 1 for abbreviations but I could not see that nay have been defined. In addition, please ensure that each table has its own caption with all abbreviations clearly defined (even if that means repeating them).

Table 3: This table is rather confusing. As presented it would suggest that anti-coags listed here are taken to prevent bleeding. This would not be the case, obviously. Column 2 header “condition” doesn’t seem appropriate as secondary prevention & acute treatment are not conditions – secondary prevention/acute treatment of what? Are there 2 rows for bleeding/primary prevention because these investigated different classes of LMWH (B01AB Vs B01AA03)? Please also see reviewer 1 comments. Please revise this table.

Please also revise the presentation of the 95% CIs in this table to ensure consistency with other tables. Suggest the use of commas rather than hyphens, as these can be confused with the reporting of negative values, to separate upper and lower bounds. Please ensure that words are not split across lines. As above please ensure an appropriate caption includes all abbreviations (e.g., DVT, PE).

DISCUSSION

Please remove all sub-headings from the discussion such that it reads as a single piece of continuous prose, ending in a one paragraph conclusion. Please present and organize the Discussion as follows: a short, clear summary of the article's findings; what the study adds to existing research and where and why the results may differ from previous research; strengths and limitations of the study; implications and next steps for research, clinical practice, and/or public policy; one-paragraph conclusion.

Please remove the funding and data availability statements for the end of the main manuscript and include only in the manuscript submission form.

Please move the ethics statement to an appropriate part of the methods section.

REFERENCES

Please ensure that for in-text reference callouts, citations are placed in square parentheses and preceding punctuation, as follows: “…is a global clinical and public health priority [1,2].” Please check and amend throughout. Please note the absence of spaces between citations.

In your bibliography, please ensure that up to but no more than 6 author names are listed, followed by et al., if more than 6 authors contribute to an individual study.

Please ensure that journal names abbreviations are those found in the National Center for Biotechnology Information (NCBI) databases

SUPPORTING INFORMATION

Please ensure that it is clearly defined for the reader what the data in supplementary tables 1 and 2 refer to. Please include an appropriate title and caption describing their contents.

Comments from the reviewers:

Reviewer #1: See attachment

Michael Dewey

Reviewer #2: This manuscript aimed to explore the impact of comorbidity on relative treatment effect in trials. This is an interesting topic. One strength of this study is that it included trials with individual patients data, and thus the assessment of effect modification is based on within-trial comparisons. However, there are several issues for the authors to address.

1. Abstract

For analysis, authors stated only they performed two-stage IPD meta-analysis. We suggest to add more details, e.g., first conducted regression analysis within trials and then combine interaction coefficients across trials for separate conditions and comparisons.

2. Abstract

In the conclusion, authors said "Our findings support the assumption that estimates of treatment efficacy are constant". We suggest to highlight that this study is about relative effect (authors have thoughtfully and appropriately discussed relative versus absolute effect in discussion), that is, this study found no evidence of relative effect differences associated with comorbidity.

3. Introduction

In the third paragraph, authors discussed limitations of subgroup analysis in meta-analysis. The authors are correct when they state: "trials rarely report subgroup effects by comorbidity, and those that do may be subject to publication bias." However, what systematic review authors do under these circumstances is utilize between-study comparisons that provide much less compelling evidence. It may be - we think it would be - worth pointing this out before moving on the merits of IPDMA. The within- versus between difference is highlighted in the first structured instrument for addressing subgroup analyses https://www.cmaj.ca/content/192/32/E901

4. Introduction

In the last paragraph, for objective of this study, authors mentioned age and sex. Although they added age and sex in the regression model, seems their primary objective is to explore effect modification by comorbidity. Authors may want to remove age and sex from the objective.

5. Methods

We suggest making it clearer about at what level (individual patients, or trials), authors extracted what data and conducted what analysis.

6. Results

To convert the measures into a similar scale, authors divided the estimates by the minimally important difference; however, this makes the interaction coefficients are difficult to interpret. We suggest adding how one should interpret the interaction coefficients for comorbidities (as they did for age treatment interactions).

7. Results

In addition to figures, better to also have tables containing the comorbidity treatment interaction coefficients, and measure of inconsistency e.g., I2 (whether interaction coefficients differ across trials). If the authors feel this is too burdensome for the main text such tables could be added in supplementary material.

8. In relation to other work the authors should discuss another study addressing subgroup effects in IPDMA

https://academic.oup.com/ije/article/48/2/596/5184552

These authors found more interactions than the current authors in a sample of IPDMA's, a couple of which were related to comorbidity. It seems to us that the results could be interpreted as essentially consistent with the current results, or somewhat inconsistent. We would be interested in the authors' views.

9. The authors deal well with the issue of multimorbidity and baseline risk. Another key contextual issue they may want to address is the burden of treatment in those with multi-morbidity who are generally on a number of medications whenever they consider adding another. Looking at some of the literature on minimally disruptive medicine may be useful in this regard. https://pubmed.ncbi.nlm.nih.gov/27417747/

Reviewer #3: Please see the attached PDF for my comments to the authors.

[LINK]

---

## [Decision Letter · Decision Letter 2]

6 Apr 2023

Dear Dr. Hanlon,

Thank you very much for re-submitting your manuscript "Treatment effect modification due to comorbidity: individual participant data meta-analyses of industry-sponsored randomised controlled trials" (PMEDICINE-D-23-00048R2) for review by PLOS Medicine.

I have discussed the paper with my colleagues and the academic editor and it was also seen again by 2 reviewers. I am pleased to say that provided the remaining editorial and production issues are dealt with we are planning to accept the paper for publication in the journal.

[LINK]

We look forward to receiving the revised manuscript by Apr 13 2023 11:59PM.   

Sincerely,

Philippa Dodd, MBBS MRCP PhD

PLOS Medicine

plosmedicine.org

Requests from Editors:

GENERAL

Thank you for your detailed and considered responses to previous editor and reviewer comments, please see further comments detailed below, which we require you address prior to publication.

INDUSTRY SPONSORED TRIALS

The editorial team agree that it would be helpful to include a clear justification for including only industry-sponsored trial data and further discussion regarding the limitations of this approach. Please see the comments from the academic editor (below) in respect of the same, which we agree with.

COMMENTS FROM THE ACADEMIC EDITOR

My initial assessment is that this is an interesting and important topic using a rather unique and hard to obtain group of data sets.

Some important caveats that the authors should emphasize are:

1) As this is industry sponsored trials, there may be some risk that these trials are not representative of the full range of trials -- designed with very tight exclusion/inclusion criteria or in other ways to boost the chance of getting a positive, significant finding in a desired direction and with very high and consistent compliance (more so than in other settings). This may compress the possibility of effect modification being detected BUT this does not mean that in other types of trials and studies the possibility of substantial effect modification still exists. Would be good to hear the authors' thoughts on this in the Discussion.

2) Null trials. If the main effect is null (or very small) it is probably more likely that there will be no (detectable) effect modification. Meta-analyzing effect modification including studies w/o a main effect may not be how we standardly think about effect modification. Would be good to hear the authors' thoughts on this in the Discussion

TITLE

Thank you for modifying your title, please remove 'industry-sponsored' suggest instead “Treatment effect modification due to comorbidity: An individual participant data meta-analysis of 126 randomised controlled trials” or something similar

INTRODUCTION

Line 131 “…byi)…” suggest “…by i)…”

TABLES

Table 2: what to the asterisks refer to, it wasn’t entirely clear upon review, apologies if I have missed it, please clearly define. Row number 4 (age-treatment interaction), to improve accessibility to the reader please ensure the upper and lower bounds of CrIs are on a single line.

Table 3: the footnote is labelled “Table 1” (line 388) please revise. Please define LMWH in the footnote.

Line 548: please remove the competing interest statement from the main manuscript and include only in the manuscript submission form when you re-submit your manuscript, it will be compiled as metadata.

REFERENCES

For intext reference callouts, please remove spaces from between different citations, for example line 89 “health priority [1, 2]…” should read “health priority [1,2]..” please check and amend throughout. 

SOCIAL MEDIA

To help us extend the reach of your research, please provide any Twitter handle(s) that would be appropriate to tag, including your own, your coauthors’, your institution, funder, or lab. Please detail any handles you wish to be included when we tweet this paper, in the manuscript submission form when you re-submit the manuscript.

Comments from Reviewers:

Reviewer #1: The authors have addressed all my points.

Just for the record, although it is not a response to me, I think not reporting p-values is fine.

Michael Dewey

Reviewer #3: The authors have addressed my concerns adequately.

[LINK]

---

## [Editor Report · Decision Letter 3]

12 Apr 2023

Dear Dr Hanlon, 

On behalf of my colleagues and the Academic Editor, Professor Jeremy Goldhaber-Fiebert, I am pleased to inform you that we have agreed to publish your manuscript "Treatment effect modification due to comorbidity: Individual participant data meta-analyses of 120 randomised controlled trials" (PMEDICINE-D-23-00048R3) in PLOS Medicine.

Before your manuscript is published, please make the following final revisions:

1) Line 470 – “However, it is possible that my selecting…” suggest “…by selecting…”

2) Line 478 – please add a space before the opening parenthesis here “…practice[35]…”

PRESS

Thank you again for submitting to PLOS Medicine, it has been a pleasure handling your manuscript. We look forward to publishing your paper. 

Best wishes, 

Pippa

Philippa Dodd, MBBS MRCP PhD 

PLOS Medicine